# Research data sharing in the Australian national science agency: Understanding the relative importance of organisational, disciplinary and domain-specific influences

Claire M. Mason[1]*, Paul J. Box[2], Shanae M. Burns[1]

1 CSIRO, Data61, Fortitude Valley, QLD, Australia, 2 CSIRO, Land & Water, Eveleigh, NSW, Australia

* Claire.Mason@data61.csiro.au

## Abstract

This study delineates the relative importance of organisational, research discipline and application domain factors in influencing researchers' data sharing practices in Australia's national scientific and industrial research agency. We surveyed 354 researchers and found that the number of data deposits made by researchers were related to the openness of the data culture and the contractual inhibitors experienced by researchers. Multi-level modelling revealed that organisational unit membership explained 10%, disciplinary membership explained 6%, and domain membership explained 4% of the variance in researchers' intentions to share research data. However, only the organisational measure of openness to data sharing explained significant unique variance in data sharing. Thus, whereas previous research has tended to focus on disciplinary influences on data sharing, this study suggests that factors operating within the organisation have the most powerful influence on researchers' data sharing practices. The research received approval from the organisation's Human Research Ethics Committee (no. 014/18).

## Introduction

Even though most researchers agree that data sharing supports scientific progress [1], actual levels of data sharing amongst researchers are relatively low [2–5]. Back in 2014, Tenopir et al. reported that fewer than 16% of the researchers made all of their data available. In spite of new policies, infrastructure and initiatives to promote research data sharing, more recent research confirms that less than one third of researchers share their data publicly [5].

Surveys exploring the barriers and enablers for sharing of research data have now been carried out in a number of research organisations, countries and disciplinary communities. These studies tell us, for instance, that researchers are less willing to share their research data if other researchers might use their data to publish before them, if they have to expend significant effort in order to share the data and if they believe that their data could be misinterpreted [4, 6–8]. While these factors represent the most immediate or proximal concerns of researchers,

**Data Availability Statement:** The raw data are stored on CSIRO's Data Access Portal. However, under the conditions of the original ethics

application, only members of the research team approved by the ethics committee are allowed to access these data. Approval would need to be granted by the ethics committee (who can be contacted at csshrec@csiro.au) for researchers seeking to verify our findings. We have published the aggregated data (for all organisational units, disciplines and domain groups with 10 or more respondents) which is allowed under the informed consent arrangements. These data are publicly available on CSIRO's Data Access Portal at https://doi.org/10.25919/5ed5e83bb35b3.

**Funding:** The author(s) received no specific funding for this work.

**Competing interests:** The authors have declared that no competing interests exist.

they reflect the institutional arrangements surrounding research practice such as the systems of rewards, rules and regulations, external legal systems and informal, epistemic community norms and conventions [9, 10]. While the digitization of data and work has made research data sharing possible, the slow take-up of research data sharing and data re-use has led to the recognition that optimal levels of data sharing will not occur without reshaping institutional arrangements so that data sharing is both facilitated and incentivized. To this end, research institutions (Universities, government, grant agencies, journal publishers) are now redesigning their infrastructure, regulation, policies and reward systems. Within government, industry and research, significant investment is being made in initiatives to support research data sharing, including for example, the adoption of FAIR data principles [11] and the CoreTrustSeal [12]. In spite of these efforts, most research data is still not publicly available [5, 7]. The failure to achieve greater impact from these initiatives may reflect the fact that the research carried out to date does not allow us to determine which institutions (those operating within research organisations, those associated with disciplinary communities or those from the application domain) have the greatest impact on researchers' data sharing practices. This study represents a first effort to answer this question using multi-level modelling. Using survey data collected from researchers working in a national science organisation, we carried out modelling to estimate the proportion of variance in researchers' intentions to share data that could be explained by their organisational unit, discipline and application domain.

## Organisational unit

Organisational work units are known to be an important source of influence on employees' attitudes and behaviours [13–15]. Data sharing research confirms that this proposition also applies to researchers' data sharing behaviour. For example, Huang et al. [16] found that amongst biodiversity scientists, respondents whose institute/university or funding agencies encouraged data sharing were more willing to share their research data and Sayogo and Pardo [8] reported that organisational involvement (in particular, organisational support for data management practices) predicted the likelihood that researchers would publish their data sets. Tenopir et al's research [7, 17] finds that the majority of respondents believe that their organisation could do more to facilitate data sharing, by providing processes, training and funding to support long term data management. Organisational factors are therefore seen to play a role in either facilitating or hindering data sharing.

## Discipline

Disciplinary influences on data sharing have also been reported by researchers and they have been estimated to explain between 19 and 5% of the variance in researchers' data sharing practices [18, 19]. Researchers who work with data from human participants (e.g., social scientists and health researchers) are known to have lower levels of data sharing than researchers from other disciplines [1] which can be attributed to the special ethical and legal requirements pertaining to human data [5, 20]. Another driver of differences between research disciplines in their data sharing practices is the distinction between 'big science' and 'small science'. Heidorn [21] argues that the large datasets collected by researchers from big science disciplines (such as astronomy, oceanography and physics) are more economical to curate than are the many small datasets captured in small science disciplines (such as biogeochemistry and psychology). Furthermore, researchers working in big science project often use the same instruments and have a greater need and motivation to coordinate efforts around data. To enable these big science initiatives, scientific infrastructure is put in place to support data storage and reuse of data [21]. In small science disciplines, research is generally driven by individual investigators

or small teams who tend to collect small scale and more heterogeneous datasets. These factors mitigate against data sharing amongst researchers from small science disciplines. Rules surrounding disciplinary data repositories, funders' policies (requiring researchers to provide a data management plan) and journal policies (requiring authors to share their data by submitting it to a data repositories) are another source of differences between disciplines which have been found to be correlated with researchers' willingness to share data [17, 22, 23]. Together, the differences in ethics requirements, the economics of data storage and sharing (in big science versus small science disciplines) and the disciplinary-specific nature of data repositories, funders' policies and journal policies will lead to disciplinary differences in data sharing practices.

## Application domain

The third potentially important source of institutional influence on data sharing practices is the researcher's application domain (or 'domain'). Whereas the researcher's discipline represents the academy or branch of knowledge that the researcher uses, the domain is the field or industry sector in which the knowledge is being applied [24]. This domain is formally classified as the field of research nominated by the researcher (e.g., in a grant application) but is also reflected in the type of organisation funding the research (e.g., a health provider vs a transport company). Each industry sector has unique legislative, regulatory and policy frameworks which establish rules and norms for data management and sharing which are reflected in the data sharing policies and IP requirements of research funders [9].

There is already evidence to suggest that these factors influence data sharing practices. While government agencies are enacting policies to mandate data sharing (because they generally fund research to inform public policy or achieve public good outcomes), private sector organisations usually fund research for private benefit and retain intellectual property rights which limit research data sharing [25]. Requirements from funding agencies to share data have been identified by researchers as important influences on their data sharing behaviour [10, 16, 22]. On the other hand, the growth in economic opportunities for commercialising data have led some industry actors to exploit new legal rights and mechanisms which allow them to maintain control over scientific data that used to be more accessible [26]. Tenopir et al's [17] study found that researchers working in different sectors (e.g., government, not-for-profit, academic, commercial) reported significantly different levels of organisational involvement, training and assistance with data management and storage. Researchers working for government were more likely to report that their organisation had processes for data management and storage and researchers working in the commercial sector were slightly less willing to share their data than researchers employed in other sectors [17].

Thus, external influences on researchers' data sharing practices can be delineated into three types: organisational, disciplinary and domain. From the surveys and interviews that have been carried out with researchers to date, we know that all three are seen to be important sources of influence on data sharing practices [7, 8, 10, 16, 22, 23]. The goal of this study was to establish the relative importance of these three sources by differentiating their impacts on research data sharing empirically.

## Methodological approach

Differentiating the effects of organisational unit, disciplinary background and research domain on data sharing behaviour is a nontrivial methodological problem. When there is dependence among observations (e.g., when individuals are subject to the same higher-level influences) standard statistical formulas will underestimate the sampling variance and therefore lead to

biased significance tests with an inflated Type I error rate. In such instances, multi-level modelling is required [27].

Two attempts to model the institutional influences on data sharing using multi-level modelling have been carried out to date but this work has focused on identifying variance in data sharing associated with the researcher's disciplinary background. Kim and Stanton [19] found that that 19% of the total variance in data sharing behaviours could be explained by researchers' disciplinary membership. A second study by Kim and Yoon [18] found that 5% of the variance could be explained by disciplinary membership (with the availability of disciplinary repositories explaining 13% of the disciplinary variance in data sharing). However, this modelling assumes that researchers with the same disciplinary background experience the same institutional influences. In practice, researchers from the same discipline may work in different application domains and (with the growth of multidisciplinary and transdisciplinary research) there may be researchers from multiple disciplines within the same organisational unit. In consequence, to understand institutional influences on data sharing we need to model disciplinary, organisational and domain factors separately.

Some combinations of discipline and domain will occur more commonly than others (e.g., astronomy researchers are not likely to work in the environmental domain) so we used partially crossed multi-level modelling [28] to estimate the relative importance of disciplinary, organisational and domain factors. This analysis allows us to quantify the proportion of variance in data sharing explained by organisational unit, discipline and domain membership. We also tested the explanatory power of specific organisational (data culture and peer encouragement), discipline (journal publishers' requirements, availability of data repositories) and domain (contractual and regulatory inhibitors) factors in explaining the variance in research data sharing. Since our goal was to delineate external influences on research data sharing, we did not introduce any individual-level (researcher) predictors in the model.

## Method

### Sample

This research was approved by the CSIRO Social and Interdisciplinary Science Human Research Ethics Committee (approval number 014/18). Participants took part in an online survey, reading the information sheet approved by the ethics committee and then clicking "I consent" to confirm that they were willing to take part in the research. Participants who selected "I do not consent" went to an exit page and did not complete the survey.

The online survey was sent to research employees at the CSIRO, a national science agency with offices across Australia. The CSIRO has characteristics in common with both Universities and government organisations. Many of CSIRO's employees have spent time working in Universities and they commonly collaborate with University researchers (e.g., in the delivery of research projects, writing of research publications and the supervision of PhD and postdoctoral students). However, whereas researchers at Universities have a teaching role, researchers in the CSIRO, an Australian federal government agency, are funded under the Science and Industry Research Act 1949 to address national objectives and support Australian industry, the community and the relevant Minister. The CSIRO is Australia's largest patent holder and CSIRO employees publish more than 3,000 refereed CSIRO journal articles and reviews per year [29]. Based on normalised citation impact, CSIRO's areas of research strength include social sciences, environment/ecology, plant and animal science, biology and biochemistry, engineering, microbiology, agricultural science, space science, and clinical medicine [29]. The CSIRO also manages infrastructure (e.g., The Australia Telescope National Facility) and biological collections (e.g., The Atlas of Living Australia, Australia's national biodiversity

database) for the benefit of research and industry. On 30 June 2017, CSIRO had a total of 5,565 staff (FTE of 4,990), of whom approximately 27% are classified as research scientists, 19% are classified as research managers and 32% are classified as research project staff [30]. Based on advice from CSIRO's Information Services team, organisational units and roles that were not likely to be dealing with research data (e.g., finance and human resources) were removed from the organisation's email list before the email containing the link to the online survey was sent out (by CSIRO's Chief Scientist) to 3,658 CSIRO employees.

Eight hundred and six employees agreed to participate, representing a 22% response rate but only 381 respondents provided sufficient data to match them to an organisational unit, discipline and application domain. The sample for the multi-level analyses was further reduced because tests of inter-rater agreement require that the number of respondents in each group should be equal to or greater than the number of response options on the Likert scales [31] and requiring that all organisational, discipline and domain groups be represented by 7 or more respondents reduced the sample size to n = 354. The gender, age, discipline and sector diversity of the survey respondents is reported in Table 1. The sample was more male dominated (81%) than the organisation as a whole (60%) but this may reflect the gender composition of employees who work with research data. The sample provided representation from a range of research disciplines (n = 12) and application domains (n = 11).

## Procedure

The survey was conducted online, with a link to the survey provided in an email from the organisation's Chief Scientist. The email explained that the survey would inform the organisation's data governance strategy and that double movie passes would be awarded to ten randomly identified survey participants. The survey was kept open for three weeks and two reminder emails were sent prior to closing the survey. CSIRO's Information Services team provided a report on the number of data deposits made to the DAP (the CSIRO's Data Access Portal) by each organisational unit. The research team integrated these data with the survey responses, using organisational unit as the linking variable.

## Measures

**Organisational unit, discipline and application domain.** Survey participants were asked to select (from a list) the option which best described the organisational unit that they worked in, their research discipline (using the Australian and New Zealand Standard Research Classification, [32]) and their application domain (using Australian Government thesaurus of government functions, [33]). Since we expected domain factors to vary depending on whether researchers were working in the public sector or the private sector, participants also specified whether they primarily worked with either Industry or Government. Thus, application domain was coded according both to the sphere in which they were operating (e.g., primary industry) and whether they were working in the public or private sector.

**Intentions to share data.** Intentions to share data were assessed by asking researchers to report the likelihood that they would share research data with a list of potential targets outside of the relevant project team (researchers in their own organisational unit, researchers in other organisational units, researchers in their own discipline, research collaborators outside of CSIRO, research funders and the general public). The items were rated on a 5-point Likert scale ranging from "Extremely unlikely" to "Extremely likely".

**Peer support.** We adapted three items from Curty's [34] measure of social influence for data re-use to create a measure of peer support for data sharing, e.g., "My peers (in CSIRO)

**Table 1. Characteristics of survey participants (n = 381).**

| Characteristics | Number of Participants |
|---|---|
| Gender | |
| Male | 242 |
| Female | 100 |
| Prefer not to answer | 11 |
| Missing | 28 |
| Age | |
| Under 25 years | 3 |
| 25 to 34 years | 30 |
| 35 to 44 years | 120 |
| 45 to 54 years | 123 |
| 55 to 64 years | 60 |
| 65 or more years | 13 |
| Missing | 32 |
| Research Discipline | |
| Mathematics | 11 |
| Physical Sciences | 22 |
| Chemical Sciences | 26 |
| Earth Sciences | 41 |
| Environmental Sciences | 106 |
| Biological Sciences | 47 |
| Agricultural and Veterinary Sciences | 43 |
| Information and Computing Sciences | 19 |
| Engineering | 27 |
| Technology | 11 |
| Medical and Health Sciences | 20 |
| Studies in Human Society | 8 |
| Application Domain | |
| Environment (private sector) | 15 |
| Environment (public sector) | 89 |
| Health Care (private sector) | 23 |
| Health Care (public sector) | 14 |
| Manufacturing (private sector) | 23 |
| Natural resources (private sector) | 42 |
| Natural resources (public sector) | 37 |
| Primary Industries (private sector) | 50 |
| Primary Industries (public sector) | 42 |
| Science (private sector) | 17 |
| Science (public sector) | 29 |

encourage me to share data". The items were rated on a 7-point Likert scale ranging from "Strongly disagree" to "Strongly agree".

 **Open data culture.** To assess shared attitudes towards data sharing within organisational units we developed six items, each reflecting the belief that data should be made as openly available as possible in order to support scientific integrity and public benefit, for example, "Open data improves scientific integrity". Each item was rated on a 7-point Likert scale ranging from "Strongly disagree" to "Strongly agree".

**Regulative pressure by journal publishers.**   Kim and Stanton's [19] four item measure was used to assess this disciplinary factor, which assesses whether or not journals require researchers to share their data when their work is published (e.g., "Journals require researchers to share data") on a 7-point Likert scale ranging from "Strongly disagree" to "Strongly agree".

**Data repositories.**   Kim and Stanton's [19] three item measure was used to assess the availability of data repositories. To ensure that the items reflected a disciplinary factor, we introduced the scale with the words "In my discipline. . ." which was followed by each question (e.g., "Data repositories are available for researchers to share data"). The items were rated on a 7-point Likert scale ranging from "Strongly disagree" to "Strongly agree". All the items were replicated from the 'Data repository' measure [19].

**Contractual and regulatory inhibitors.**   To assess the impact of domain factors on data sharing, we asked researchers to rate the extent to which factors in their industry sector or government area inhibited their ability to share data, using a 7-point Likert scale ranging from "Not at all" to "A great deal". A principal component analysis carried out on the five items revealed that it formed two separate factors. We labelled the first factor contractual inhibitors (e.g., "Contractual conditions i.e., the terms of the contract under which the data were generated or used") and the second factor regulatory inhibitors (e.g., "Privacy requirements").

**Data deposits in the organisational repository.**   The CSIRO has an organisational repository known as the Data Access Portal which provides a secure mechanism for depositing data and software code. When employees publish their data on the platform, they are required to report which organisational unit they work in. We obtained a report which allowed us to count how many collections had been published on the portal for each organisational unit and we linked this measure with the survey data. However, the measure was highly positively skewed (many organisational units had either no deposits or very few deposits whereas a small number of units made very frequent deposits). Since extreme scores can have too much impact in analyses, we converted the measure of data deposits into a categorical variable (organisational units were classified as either having no deposits, fewer than five deposits, five to nine deposits or ten or more deposits).

## Statistical analysis

The statistical modelling was carried out in R [35]. The multilevel package [36] was used for tests of within-group agreement and reliability while the lme4 package [37] was used to test the multi-level model since it is particularly well-suited to dealing with partially crossed data structures [38]. Since the data were not balanced (not all combinations of organisational unit, disciplinary and domain membership were represented in the data) we fitted our models using restricted maximum likelihood estimation (REML), a modification of maximum likelihood estimation that is more precise for mixed-effects modelling. However, when comparing the fit of alternative models it was necessary to use the standard maximum likelihood estimation. We used the ANOVA function in lmerTest [39] to obtain provide p-values for each explanatory factor in the multi-level model since Luke [40] reports that the Kenward-Roger and Satterthwaite approximations provided produce acceptable Type 1 error rates even for smaller samples. For all statistical procedures, an alpha level of 0.05 (two-tailed) was used to determine statistical significance.

Prior to carrying out our analyses we cleaned the data, checking that the data were normally distributed (as noted above, the measure of data deposits was highly skewed and was converted to a categorical variable) and removing records of participants who (a) did not specify which organisational unit, discipline and domain they worked in or (b) who came from organisational units where none of the respondents reported working with research data. This gave us

a sample of 381 respondents and 31 organisational units for the initial analyses (factor analysis and correlations). The sample size for the multi-level analyses was further reduced because we removed data from organisational units, disciplines and domains that were represented by fewer than seven respondents. This left us with survey data from 354 researchers, who collectively represented 28 organisational units, 12 disciplinary groups and 11 application domains.

Additional decisions regarding our statistical procedures are described in the results section, as they emerged in the course of our analyses.

## Results

Before commencing the multi-level modelling, we performed a principal component analysis to check the construct validity of the measurement items. The solution suggested extracting seven factors from the data and when we ran the principal component factor analysis with varimax rotation the seven factors explained 73% of the variance. The rotated factor solution exhibited good simple structure, with all items loading above 0.62 on their intended construct and none loading above 0.27 on other factors (see Table 2). Alpha coefficients for each scale are reported in the diagonal of Table 3. All measures displayed satisfactory reliability and validity (alpha coefficients of 0.70 or higher).

We also aggregated these measures to the organisational unit level so that we could check whether they were correlated with the real-world measure of data sharing (number of deposits on the organisational data repository, classified as either none, fewer than 5, five to nine deposits or ten or more deposits). The correlations among the survey measures and the categorical measure of data deposits are shown in Table 3. We found that openness of the data culture (r = 0.39, p <0.05) and contractual inhibitors (r = -0.38, p <0.05) correlated significantly with organisational unit deposits. Regulatory inhibitors were marginally significantly correlated with data deposits (r = -0.35, p <0.10) but intentions to share data were not significantly correlated with data deposits (r = 0.21, p >0.05) and nor were the disciplinary measures (journals and repositories).

### Estimating within-group agreement and reliability

Before carrying out multi-level analyses it is necessary to check whether the measures exhibit within-group agreement and between-group variance. If researchers from the same discipline provide similar ratings when asked about the level of regulatory pressure from journals but differ in their ratings when compared to researchers from other disciplines it supports treating the measure as a construct pertaining to the research discipline. To assess the level of within-group agreement, we calculated the multi-item $r_{wg(j)}$ statistic [41]. By convention, values at or above 0.70 are considered good agreement [38] but we also tested the statistical significance of the $r_{wg(j)}$ values by simulating rwg(j) values from a uniform null distribution for user supplied values of (a) average group size, (b) number of items in the scale, and (c) number of response options on the items. The results of these tests (see Table 4) indicated that there was greater agreement within organisational unit, disciplinary and domain groups for the relevant organisational, disciplinary and domain factors than would be expected by chance.

Interclass correlations (ICCs) were calculated to check that each measure had significant between-group variance (see Table 4). The ICC(1) statistic represents the proportion of variance in the measure which is explained by the grouping factor whereas the ICC(2) represents another way of measuring agreement in group members' ratings of the constructs. According to James [42], the median ICC(1) reported for group-level constructs is 0.12 and values between 0.05 and 0.20 are acceptable. All but the measure of regulative pressure from journals met this standard. The ICC(2) values were also acceptable for all measures except regulative pressure by journal publishers and peer support (values above 0.70 are generally agreed to

**Table 2. Pattern matrix generated from the principal components analysis.**

| Item | F1 | F2 | F3 | F4 | F5 | F6 | F7 | h² |
|---|---|---|---|---|---|---|---|---|
| The following statements represent alternative ways of thinking about the role of data in the CSIRO. Please choose the response option which best describes your level of agreement with this statement. | | | | | | | | |
| Data should be open and accessible by default (access to data should only be restricted where privacy, confidentiality or IP issues require it) | 0.73 | 0.22 | 0.11 | 0.02 | 0.14 | -0.07 | 0.07 | 0.63 |
| Our data should be harnessed for public good | 0.71 | 0.15 | 0.01 | 0.13 | -0.02 | -0.07 | -0.06 | 0.55 |
| Data sharing can have global and intergenerational benefits | 0.73 | 0.24 | 0.02 | 0.06 | 0.07 | -0.10 | 0.03 | 0.61 |
| The integrity of our research is improved when our data are available for others to review | 0.84 | 0.10 | 0.07 | 0.07 | -0.03 | 0.01 | -0.09 | 0.74 |
| Open data improves scientific integrity | 0.87 | 0.03 | 0.06 | 0.04 | 0.08 | 0.04 | -0.07 | 0.78 |
| Making our data more accessible will improve the quality of our research | 0.81 | 0.11 | 0.07 | 0.02 | 0.02 | -0.05 | 0.01 | 0.67 |
| In the next 12 months, how likely is it that you will share data from one of your research projects with: | | | | | | | | |
| Colleagues in my business unit | 0.06 | 0.66 | 0.08 | 0.27 | 0.03 | 0.20 | -0.16 | 0.59 |
| Colleagues in other business units | 0.17 | 0.74 | -0.03 | 0.12 | 0.07 | 0.15 | 0.03 | 0.62 |
| Colleagues in my research discipline | 0.18 | 0.82 | 0.12 | 0.13 | 0.11 | -0.18 | -0.12 | 0.80 |
| Other researchers | 0.16 | 0.80 | 0.10 | 0.11 | 0.14 | -0.13 | -0.10 | 0.73 |
| Research collaborators/partners outside of CSIRO | 0.10 | 0.70 | 0.05 | 0.05 | 0.06 | -0.17 | -0.06 | 0.54 |
| General public | 0.25 | 0.64 | 0.10 | 0.02 | 0.12 | -0.20 | -0.04 | 0.54 |
| In my discipline… | | | | | | | | |
| Data sharing is mandated by journals' policy | 0.11 | 0.10 | 0.87 | 0.10 | 0.20 | -0.01 | -0.04 | 0.83 |
| Data sharing policy of journals is enforced | 0.06 | 0.08 | 0.89 | 0.09 | 0.14 | 0.02 | 0.03 | 0.83 |
| Journals require researchers to share data | 0.07 | 0.10 | 0.92 | 0.01 | 0.13 | -0.02 | -0.05 | 0.87 |
| Journals can penalize researchers if they do not share data | 0.06 | 0.04 | 0.83 | 0.00 | 0.01 | -0.07 | 0.00 | 0.69 |
| In my opinion, my peers (in CSIRO)… | | | | | | | | |
| Encourage me to share data | 0.10 | 0.17 | 0.03 | 0.89 | 0.11 | -0.06 | -0.07 | 0.85 |
| Are supportive of the sharing of data | 0.14 | 0.14 | 0.06 | 0.90 | 0.09 | -0.08 | -0.07 | 0.87 |
| Often share data | 0.05 | 0.18 | 0.09 | 0.86 | 0.12 | 0.00 | -0.07 | 0.81 |
| In my discipline… | | | | | | | | |
| Researchers can easily access data repositories | 0.08 | 0.16 | 0.18 | 0.12 | 0.85 | -0.05 | -0.07 | 0.81 |
| Data repositories are available for researchers to share data | 0.11 | 0.16 | 0.13 | 0.12 | 0.89 | -0.07 | -0.06 | 0.86 |
| Researchers have the data repositories necessary to share data | 0.01 | 0.09 | 0.15 | 0.08 | 0.89 | -0.04 | -0.07 | 0.84 |
| Within this area, how much do each of the following factors inhibit your ability to share data? | | | | | | | | |
| Contractual conditions i.e., the terms of the contract under which the data were generated or used | -0.10 | -0.17 | -0.08 | -0.07 | -0.03 | 0.84 | 0.17 | 0.79 |
| Constraints imposed by the ownership or licensing arrangements of third party data | -0.05 | -0.02 | 0.04 | -0.04 | -0.06 | 0.85 | 0.16 | 0.76 |
| Ethical restrictions | 0.05 | -0.14 | -0.01 | -0.18 | -0.11 | -0.02 | 0.84 | 0.78 |
| Privacy requirements | -0.07 | -0.21 | -0.08 | -0.14 | -0.09 | 0.47 | 0.63 | 0.70 |
| Other legislation, regulation and policy (e.g., anti-trust concerns) | -0.10 | -0.08 | 0.01 | 0.07 | -0.05 | 0.29 | 0.75 | 0.67 |

Alpha coefficients for each scale are shaded grey.

represent sufficiently high agreement to support aggregation, [43]). Based on the low intraclass correlations for the measure of regulatory pressure by journal publishers, we did not include this measure in the multi-level modelling. We retained the measure of peer support since it demonstrated acceptable within-group agreement on the $r_{wg(j)}$ statistic and acceptable between-group variance.

## Testing the random effects model

The first step in building a multi-level model is to estimate the random effects model (in which there are no predictors but there is a random intercept variance term for the different grouping

**Table 3. Organisational unit-level correlations among survey measures and data deposits (N = 31).**

| | Mean (SD) | 1 | 2 | 3 | 4 | 5 | 6 | 7 |
|---|---|---|---|---|---|---|---|---|
| 1. Intentions to share data | 3.90 (0.59) | 0.86 | | | | | | |
| 2. Peer support | 4.39 (0.50) | 0.56** | 0.91 | | | | | |
| 3. Open data culture | 3.46 (0.77) | 0.21 | 0.15 | 0.89 | | | | |
| 4. Journals | 3.85 (0.74) | 0.36† | 0.25 | 0.48** | 0.92 | | | |
| 5. Repositories | 3.67 (0.49) | 0.15 | -0.06 | 0.50** | 0.43* | 0.90 | | |
| 6. Contractual inhibitors | 4.21 (0.49) | 0.03 | 0.17 | -0.44* | -0.03 | -0.19 | 0.79 | |
| 7. Regulatory inhibitors | 1.90 (0.42) | -0.11 | -0.19 | 0.01 | -0.08 | -0.07 | 0.41* | 0.73 |
| 8. Data deposits | 1.58 (0.46) | 0.21 | 0.08 | 0.39* | 0.03 | 0.07 | -0.38* | -0.35† |

* p < 0.05

** p < 0.01.

† p < 0.10.

Standardized alpha coefficients for each survey measure are reported in the diagonal.

variables and all combinations thereof). The model explains intentions to share data of researcher $i$ in organisational unit $j$, and discipline $k$ and domain $l$ as follows:

$$Y_{ijkl} = \gamma_0 + u_j + v_k + w_l + x_{jk} + y_{jl} + z_{kl} + a_{jkl} + e_{ijkl}$$

There are eight random effects in Equation (1):

$u_j \sim N(0,\sigma^2_u)$ which is the random effect of organisational unit $j$

$v_k \sim N(0,\sigma^2_v)$ which is the random effect of discipline $k$

$w_l \sim (0,\sigma^2_w)$ which is the random effect of domain $l$

$x_{jk} \sim N(0,\sigma^2$ which is the random effect of the interaction between organisational unit $j$ and discipline $k$

$y_{jl} \sim N(0,\sigma^2_y)$ which is the random effect of the interaction between organisational unit $j$ and domain $l$

$z_{kl} \sim N(0,\sigma^2_z)$ which is the random effect of the interaction between discipline $k$ and domain $l$

**Table 4. Mean $r_{wg(j)}$ values and intraclass correlations for survey measures (n = 354).**

| Variable | $r_{wg(j)}$ | ICC(1) | ICC(2) |
|---|---|---|---|
| Organisational unit groups | | | |
| Intentions to share data | 0.7609** | 0.1916 | 0.7634 |
| Peer support | 0.6795* | 0.1165 | 0.6421 |
| Openness | 0.8834** | 0.1680 | 0.7331 |
| Disciplinary groups | | | |
| Intentions to share data | 0.7096** | 0.1234 | 0.8171 |
| Journals | 0.8115** | 0.0404 | 0.5722 |
| Repositories | 0.7368** | 0.1185 | 0.8102 |
| Domain groups | | | |
| Intentions to share data | 0.7264** | 0.1063 | 0.8048 |
| Regulatory inhibitors | 0.5838** | 0.1444 | 0.8539 |
| Contractual inhibitors | 0.5602** | 0.2295 | 0.9116 |

* Denotes that the rwg(j) value is above the upper expected 95% confidence interval estimated using rwg.j.sim.

** Denotes that the rwg(j) value is above the upper expected 99% confidence interval estimated using rwg.j.sim.

$a_{jkl} \sim N(0,\sigma^2_a)$ which is the random effect of the three-way interaction between organisational unit $j$, discipline $k$ and domain $l$, and

$e_{ijkl} \sim N(0, \sigma^2_e)$ which is the random effect of researcher $i$ in organisational unit $j$ and discipline $k$ and domain $l$.

This model provides estimates of the variability in the intercepts, or in other words, the proportion of variance in intentions to share data that is associated with organisational unit, research discipline and application domain (and potentially all possible combinations of these variables).

While it is generally recommended that the possibility of interactions between crossed factors should be tested in cross-classified random effects modelling [44], it is also well known that complex models incorporating all possible interactions often fail to converge [45]. When we attempted to test the 'maximal' full random effects model, it had singular fit, a common outcome when the random effects structure is too complex for the data. In such cases, if there is no theoretical reason for expecting a random effect to be significant, the most complex element should be removed from the model [45]. Following these guidelines, we tested the random effects model again, gradually removing the most complex elements (first the three-way interaction, then the organisational unit by discipline interaction and finally the discipline by domain interaction term). At this point (with the three random effects of interest and a significant organisational unit by domain interaction included) the model converged.

The resulting model included the three random effects of interest (organisational unit, domain and discipline) and an organisational unit by domain interaction. The organisational unit by domain interaction was not of theoretical interest so we tested whether the fit of the model worsened when the organisational unit by domain interaction was removed (filtering the dataset to ensure that there were at least five respondents for each unique organisational unit and domain combination) and we found that it did not ($\chi^2 = 0.8392$, p = 0.36). With three random effects remaining in the model, we then checked whether the fit of the model worsened if the random effect for domain (which explained the least variance) was removed. This test revealed that model fit worsened significantly when the random effect for domain was not included ($\chi^2 = 4.38$, p $<0.05$). Importantly, this test supports our hypothesis that data sharing reflects the combined effect of individual, organisational unit, discipline and domain influences.

The output from this random effects model is presented in Table 5 below. We can calculate the proportion of variance explained by each grouping factor by dividing the corresponding variance component by the total of all variance components in the model. For example, the variance in intercepts for organisational unit membership is 0.07986 which represents 10.42% of the total variance (0.07986 + 0.04697 + 0.03048 + 0.60909) in intentions to share data.

**Table 5. Random effects model explaining intentions to share data (n = 354[†]).**

| Random effects | Name | Variance | Std.Dev. |
|---|---|---|---|
| Organisational unit | (Intercept) | 0.07986 | 0.2826 |
| Discipline | (Intercept) | 0.04697 | 0.2167 |
| Domain | (Intercept) | 0.03048 | 0.1746 |
| Residual | | 0.60909 | 0.7804 |
| **Fixed effects** | **Coefficient** | **SE** | **t ratio** |
| Data sharing ($\gamma_{00}$) | 3.4978 | 0.1114 | 31.41*** |

*** p $<0.001$.

† The 354 researchers represented in the dataset represented 28 organisational units, 12 research disciplines and 11 application domains.

Organisational unit membership is therefore most important since it explains 10.42% of the total variance in intentions to share data. Disciplinary membership explains 6.13% of variance in data sharing and domain membership explains 3.98% of the variance in data sharing. The remaining 79.47% of variance is residual variance, attributable to either individual researcher factors or error. It is worth noting the difference between these variance estimates and the intraclass correlations (ICC(1)) reported earlier (19%, 12% and 11% respectively). The intra-class correlations overestimate the variance attributable to each grouping factor because they do not take into account the effects due to the other (correlated) grouping factors. Similarly, Kim and Stanton's higher estimate of variance due to disciplinary factors (19.1%) is probably inflated because they did not model other grouping factors (university or departmental membership, domain membership) that would have contributed to non-independence in their data.

## Testing the full model

The next step involves testing the full model in which the explanatory variables (open data culture, peer support, data repositories, contractual inhibitors and regulatory inhibitors) are tested as predictors of organisational-, disciplinary- and domain-specific variance in intentions to share data.

With the explanatory variables in the model, the total unexplained variance in the model was reduced from 0.7664 (random effects model) to 0.6763 (full model), indicating that the organisational unit, disciplinary and domain factors explained 12% of the variance in data sharing (see Table 6). However, only one of the predictors explained significant unique variance in data sharing. The openness of the data culture (organisational unit members agreeing that data sharing can have global and intergenerational benefits and that data sharing supports scientific integrity) was a significant predictor of intentions to share data (t = 2.19, p < 0.05). None of the other factors in the model explained significant unique variance.

## Discussion

The goal of this research was to delineate the relative importance of organisational, disciplinary and domain effects on research data sharing. To date, these factors have not been clearly differentiated and in consequence, estimates of the importance of different variables pertaining to

**Table 6. Full model explaining intentions to share data (n = 354[†]).**

| Random effects | Variance | Std.Dev. | |
|---|---|---|---|
| Organisational unit | 0.0255 | 0.1598 | |
| Discipline | 0.0171 | 0.1308 | |
| Domain | 0.0157 | 0.1253 | |
| Residual | 0.6180 | 0.7861 | |
| **Fixed effects** | **Beta** | **Standard error** | **t** |
| Data sharing ($\gamma_{00}$) | 0.7818 | 1.0735 | 0.73 |
| Peer support | 0.1529 | 0.1389 | 1.10 |
| Openness | 0.3363 | 0.1534 | 2.19* |
| Data repositories | 0.1947 | 0.1293 | 1.51 |
| Contractual inhibitors | -0.0447 | 0.1049 | -0.42 |
| Regulatory inhibitors | -0.1982 | 0.1511 | -1.31 |

* p < 0.05.

† The 354 researchers represented in the dataset represented 28 organisational units, 12 research disciplines and 11 application domains.

these factors (such as disciplinary norms or organisational rewards or contractual conditions) are likely to have been biased. By using a multilevel modelling technique which differentiates these influences statistically, we found that there is an independent effect from organisational unit, research discipline and application domain on researchers' intentions to share data. Furthermore, whereas previous research has tended to focus on the role of disciplinary norms and resources in influencing data sharing practices [e.g., 18, 19, 46, 47], this study suggests that factors operating at the organisational unit level have the most powerful influence on researchers' data sharing practices.

This study is also the first to demonstrate that self-report measures pertaining to data sharing are correlated with real-world data sharing behaviour. The measure of intentions to share data was not correlated with organisational units' data deposits but this finding is likely to reflect the fact that researchers share their data via a wide range of channels (the researchers in our survey reported using emails, ftp sites, Dropbox and internal shared drives to share their research data). In light of the sample size for this analysis (n = 31) it was extremely encouraging to find that some of the survey measures (namely, openness of data culture and contractual inhibitors) were correlated with this real-world data sharing behaviour.

The findings from the random effects model (which revealed that researchers' organisational unit, disciplinary and domain membership each explain unique variance in intentions to share data) are just as important as the findings from the full model. The statistical power of multi-level models is influenced both by the number of groups in the sample and the number of individuals in each group [48] making it especially challenging to achieve high statistical power when assessing effects for multiple, crossed institutional factors. The fact that open data culture emerged as a significant predictor not only reflects the fact that organisational unit explains more of the variance in data sharing behaviour but also the larger number of organisational units represented in our sample. Power in a multilevel analysis reflects the number of observations at the level of the effect being detected [49] and our sample provided observations from 28 organisational units but only 12 research disciplines and 11 application domains. Therefore, the failure to observe significant effects for the disciplinary and domain variables in the model may be due to low statistical power and we recommend retaining these factors for further investigation with a larger sample.

## Limitations

Some limitations associated with this study should be acknowledged before considering the implications of the research. Since we collected data from only one organisation, we were not able to investigate how much influence between-organisation factors have on research data sharing. In addition, we were not able to model the full organisational structure. In the organisation where this study was conducted, researchers are structured within Business Units, Research Programs and Research Groups. We carried out our analyses on Programs because our initial analyses indicated that little additional variance was explained by Business Units and Groups. Even with these compromises, the power for our analyses was low.

Second, as is common in surveys of this nature [19, 22], the survey had a low response rate (22%) which means that the responses may not have been representative of all researchers in the organisation. Perhaps more important, our model was tested in a government research organisation. Researchers working in government organisations may experience less autonomy than researchers in Universities, who are generally free to direct their own research agenda. Researchers in government organisations may also be more likely to work in multidisciplinary organisational units than researchers in a university setting. Both factors might explain why the organisational unit emerged as such an important predictor of variance in

data sharing in this study. Furthermore, government employees may be under less pressure to publish in academic journals, making the effect of disciplinary factors less important. Nevertheless, we believe it has been useful to explore institutional factors affecting research data sharing in a government science organisation, since most research on this subject has focused on researchers employed in Universities. Future research is needed to explore whether our findings generalise to other organisations and settings.

## Practical implications

This study was intended to provide insights that could be used to guide the design of interventions to facilitate research data sharing within CSIRO. The fact that we were able to differentiate the effect of discipline, organisational unit and domain membership on intentions to share data suggests that a whole of ecosystem approach will be needed to achieve *optimal* levels of data sharing. However, of the three sources of influence that we investigated, it was those within the organisation (i.e., organisational unit membership) which were most strongly related to intentions to share data. This understanding has some important implications for those seeking to improve organisational approaches to maximising the utility of data resources.

First, it suggests that improvements in data sharing practices can be achieved within organisational units, without having to rely on or influence change in external organisations or institutions (e.g. clients, academies or professional bodies). Second, it suggests that a one size fits all approach to improving data sharing within an organisation is not likely to be most effective. Instead, initiatives should be co-designed with researchers since they need to reflect local conditions and work practices. Fortunately, research suggests that there are multiple levers which organisations units can choose from to support data sharing, such as training, rewards, policies and infrastructure and services to support data management [1, 7, 8, 17, 22, 50, 51].

Second, our findings point to the importance of culture (specifically, shared beliefs about the public and scientific value of data sharing) as a driver of data sharing practices. Having an open data culture was correlated with the real-world measure of data sharing (deposits in the organisation's data repository) and explained significant unique variance in researchers' intentions to share data. This finding suggests that initiatives to support sharing are likely to be more successful when they emphasize the intrinsic benefits of data sharing (scientific integrity and public benefit) rather than extrinsic reasons for sharing data (such as funder requirements or organisational efficiencies). Hard interventions (such as rules, rewards and policies) may serve as a signal which helps to shape the data culture [51] but they should not crowd out intrinsic motivations to support data sharing. The importance of intrinsic motivation for data sharing has been found in other studies besides this one. For example, Brooks, Heidorn, Stahlman and Chong [52] found that researchers emphasize common good and the potential for transformative science when explaining their efforts to support data sharing in the context of institutionalized pressures and economic pressures constraining data sharing.

## Theoretical implications and future directions

This study replicates and extends Kim and Stanton's [19] efforts to model the role of institutional factors in influencing researchers' data sharing. Not only did we replicate their finding that researchers' disciplinary backgrounds can explain variance in their intentions to share data, this variance could be differentiated from that explained by organisational-unit and domain. Our study also extends prior research by testing these factors as a driver of data sharing in a non-traditional research organisation (i.e., not in a university setting) and

demonstrating that self-report measures pertaining to data sharing (data culture and lack of contractual inhibitors) are correlated with real world data sharing.

Data culture appears to be an especially important determinant of research data sharing. Culture reflects a shared view on 'how we do things around here' and because it reflects taken-for-granted assumptions and norms it tends to be good at predicting discretionary behaviours (such as data sharing). However, our findings are based on research carried out within one organisation. Further research is needed both to test the generalisability of our findings and to determine whether data culture is most powerful at the organisational unit level or whether between-organisation differences in data culture also influence data sharing practices. Exploring data culture across organisations may reveal other dimensions of data culture (e.g., risk-avoidance) that are relevant for data practices.

## Conclusion

Research data sharing is important because of the scientific and broader public benefits which flow from this behaviour. However, it is also of interest because of the challenges associated with inducing researchers to invest personal effort towards sharing data (so that its inherent value can be realised) when the benefits flow to others (other researchers, society, future generations, [53]). In such contexts, it is appropriate to consider how organisational, disciplinary and domain factors can be utilised to facilitate the desired behaviour. However, ultimately, shared beliefs and values within the researcher's local work environment may be most influential in shaping this socially-valued outcome.

## Supporting information

**S1 File. Discipline aggregation.**
(XLSX)

**S2 File. Domain aggregation.**
(XLSX)

**S3 File. Orgunit aggregation.**
(XLSX)

## Author Contributions

**Conceptualization:** Claire M. Mason, Paul J. Box.

**Data curation:** Claire M. Mason, Shanae M. Burns.

**Formal analysis:** Claire M. Mason.

**Project administration:** Shanae M. Burns.

**Writing – original draft:** Claire M. Mason.

**Writing – review & editing:** Paul J. Box, Shanae M. Burns.

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
