## [Decision Letter · Decision Letter 0]

16 Apr 2020

PONE-D-20-07169

Modelling organisational, disciplinary and domain-specific sources of influence in research data sharing

PLOS ONE

Dear Dr Mason,

Thank you for submitting your manuscript to PLOS ONE. After careful consideration, we feel that it has merit but does not fully meet PLOS ONE’s publication criteria as it currently stands. Therefore, we invite you to submit a revised version of the manuscript that addresses the points raised during the review process.

Both reviewers made valuable comments on the manuscript, and I'm advising you to include as much as possible in the revised version. The title of the manuscript needs to be more focused, the paper rationale and objectives should be more clear and in line with reported results. The methodology section needs an extensive revision, as some analyses are not reported or not clear, and the sampling is not well explained to the readers. As the reviewers pointed out, data behind the analyses are not available for readers. Please include the dataset in the next version, e.g., by anonymizing the data. If not possible, please be more explicit about why this is the case.

We would appreciate receiving your revised manuscript by May 31 2020 11:59PM. To enhance the reproducibility of your results, we recommend that if applicable you deposit your laboratory protocols in protocols.io, where a protocol can be assigned its own identifier (DOI) such that it can be cited independently in the future. For instructions see: http://journals.plos.org/plosone/s/submission-guidelines#loc-laboratory-protocols

We look forward to receiving your revised manuscript.

Kind regards,

Laurentiu Rozylowicz, Ph.D.

Academic Editor

PLOS ONE

Journal Requirements:

Reviewers' comments:

Reviewer's Responses to Questions

**Comments to the Author**

1. Is the manuscript technically sound, and do the data support the conclusions?

Reviewer #1: Partly

Reviewer #2: No

2. Has the statistical analysis been performed appropriately and rigorously? 

Reviewer #1: Yes

Reviewer #2: I Don't Know

3. Have the authors made all data underlying the findings in their manuscript fully available?

Reviewer #1: No

Reviewer #2: No

4. Is the manuscript presented in an intelligible fashion and written in standard English?

Reviewer #1: Yes

Reviewer #2: Yes

5. Review Comments to the Author

Reviewer #1: The manuscript addresses a relevant research topic for nowadays science: data-sharing practices. Overall, the manuscript is presented in an intelligible fashion and written in standard English, while the statistical analysis was performed appropriately and rigorously. Even if the manuscript has merits, there are several aspects that need authors attention and impede the potential publication.

I would start by stressing the irony surrounding this paper: comparing the results reported by the authors and their justification referring to the availability of the data-set supporting this manuscript.

#1. The title of the manuscript is too general (e.g. it should stress that it is a research conducted on an Australian governmental agency). From a certain perspective, some would argue that it could potentially mislead readers.

#2. The problem or the rationale of the paper is clouded by the diffuse writing. It should clearly answer "why the effort?" and also the contribution to the field.

#3. The abstract contains extremely detailed information on the results (e.g. that organizational unit membership explained 10%, disciplinary membership explained 6%, and domain membership explained 4% of the variance in researchers’ intentions to share research data) and ignores for instance the limits, the contribution to the field, etc. Additionally, it refers to specific concepts that cannot be understood appropriately unless the entire content is read (e.g. contractual and regulatory inhibitors).

#4. The theory section does not support the variables introduced into the statistical models. Recent literature on the topic is missing. I would suggest the authors to re-shape and re-frame the theory part of their paper.

#5. The authors do not explain why academic and non-academic data are equivalent (see section "Application domain").

#6. It is safe to delete Figure 1 - it is redundant, it does not provide any useful information to the readers.

#7. Background information on the respondents is missing (publication records, research impact, tenure, experience etc.). Also, it is not self-evident how representative are respondents for the Australian research community.

#8. Paragraph between lines 170 - 180 is not clear. Authors should make clear their point there.

#9. Authors should make their methodology clear. See the below questions for illustration:

- How were the employees recruited? Why CSIRO? Is it relevant for Australia?

- What is the total number of employees at CSIRO? Did the authors send emails to everybody? What is the meaning of 22%? Is the sample relevant for the entire agency? But for the entire Australia?

- Authors should explain the methodological limits for this paper's results and findings in the context of working with a convenience sample.

- Some of the disciplines are better represented in the sample. Does that impact upon the results? What is the structure of the agency's workforce on these research disciplines and application domains?

- Lines 200 - 205: How many were there? What were the criteria applied by the Information Services expert on selecting the "active research roles". Did he do a good job, given that some of the respondents reported 0 time spent on research data?

- Some of the text in the Methods sections is redundant or common knowledge, e.g. "The statistical modelling was carried out using R [29], an opensource language and environment for statistical computing."

- Text between lines: 298 - 305 should be moved somewhere in the above section, as it does not report any results. It is only technical procedure.

#10. Some parts of the text in the discussion section are not supported by the results, e.g. "The failure to observe significant effects for the other variables in the model may simple reflect low statistical power."

#11. What is the share of research productivity in Australia accounted for by the government organizations and not universities? Authors do not provide any context information.

Reviewer #2: This paper considers whether organisational units and application domains could influence data sharing practices among researchers. This is clearly a very important issue as data sharing can elevate levels of transparency in scientific endeavours as well as even increase the overall trust and confidence in scientific methods and conclusions. I applaud authors’ efforts to investigate this important question in a unique sample of non-academic researchers.

I must, however, admit that I find it a bit ironic that a manuscript discussing data sharing practices is submitted for publication without openly shared data. I have noted that this is due to ethics application stating that data would not be shared unless it is for validation. Could the data be uploaded to a password protected website with access given to researchers when necessary? I think this could be a good compromise between ethical restrictions and ensuring that data is ready to be shared when needed. If this is not possible, would authors be willing to at least share their R code along with the outputs?

Personally, I do not find the paragraph regarding the influence of discipline on data sharing too convincing. Authors argue that data sharing is more common in “big science” disciplines than “small science” disciplines. By default, data would be shared within teams with bigger teams being associated with data being shared to a bigger group of people. It would be helpful if authors could describe at least one study in detail which supports the argument that small science disciplines are less likely to share data than big science disciplines, beyond referring to explanations relating to norms. The analysis conducted by authors do not shed more light on whether this argument is true or not. In this case, perhaps it would be better to focus on how norms within disciplines can influence data sharing practices.

In lines 221-228, intentions to share data, authors report that all items loaded on a single factor, yet no factor analysis is reported in the manuscript. I am somewhat surprised that authors would treat these as a single factor especially there may be different motivations for sharing data within own organisational unit (e.g., collaboration) vs research funders (e.g., requirement) vs general public (e.g., seeing data sharing as important). Alpha of 0.71 may be reflective of this. Have authors considered treating these items as separate outcome variables? Factor analysis should also be reported for contractual and regulatory inhibitors (relating to the domain influences).

Finally, I cannot comment much on the analytic approach as I have not been trained in multilevel modelling, but could authors report on what sample sizes would be required to reach an adequate power (i.e., 80% or above) in the current analysis? Also, I am not sure why values relating to explained variance in lines 405-408 do not match those in Table 4.

6. PLOS authors have the option to publish the peer review history of their article (what does this mean?). If published, this will include your full peer review and any attached files.

Reviewer #1: No

Reviewer #2: No

---

## [Author Response · Author response to Decision Letter 0]

22 Jun 2020

Thank you to all for your helpful feedback. Our detailed responses are in the attached cover letter.

---

## [Decision Letter · Decision Letter 1]

24 Jul 2020

PONE-D-20-07169R1

Research data sharing in the Australian national science agency: Understanding the relative importance of organisational, disciplinary and domain-specific influences

PLOS ONE

Dear Dr. Mason,

Thank you for submitting your manuscript to PLOS ONE. After careful consideration, we feel that it has merit but does not fully meet PLOS ONE’s publication criteria as it currently stands. Therefore, we invite you to submit a revised version of the manuscript that addresses the points raised during the review process.

The paper is much improved and it can be accepted. However, there are some formating issues who needs to be considered during a quick revision:

- check if the manuscript is formatted according to PLOS ONE Submission Guidelines (see instructions for authors https://journals.plos.org/plosone/s/submission-guidelines). Please pay attention to tables (see the instructions here https://journals.plos.org/plosone/s/tables)

- please use leading zero in the paper (0.27 and not .27 see, e.g. ln 319 but also in many other places, including some tables)

- please proofread the paper to be sure everything is in the right place and correct.

We look forward to receiving your revised manuscript.

Kind regards,

Laurentiu Rozylowicz, Ph.D.

Academic Editor

PLOS ONE

Reviewers' comments:

Reviewer's Responses to Questions

**Comments to the Author**

1. If the authors have adequately addressed your comments raised in a previous round of review and you feel that this manuscript is now acceptable for publication, you may indicate that here to bypass the “Comments to the Author” section, enter your conflict of interest statement in the “Confidential to Editor” section, and submit your "Accept" recommendation.

Reviewer #1: (No Response)

Reviewer #2: All comments have been addressed

2. Is the manuscript technically sound, and do the data support the conclusions?

Reviewer #1: Yes

Reviewer #2: Yes

3. Has the statistical analysis been performed appropriately and rigorously? 

Reviewer #1: Yes

Reviewer #2: I Don't Know

4. Have the authors made all data underlying the findings in their manuscript fully available?

Reviewer #1: No

Reviewer #2: No

5. Is the manuscript presented in an intelligible fashion and written in standard English?

Reviewer #1: Yes

Reviewer #2: Yes

6. Review Comments to the Author

Reviewer #1: I thank the authors for the time invested in improving their manuscript. They have made a decent job in addressing the previous comments I made (except for the *data availability* remarks). Consequently, on my part, I am satisfied with the current form of the manuscript.

However, I would *stress* a few rather formal aspects that the authors should seriously take into consideration prior potential publishing their manuscript.

#1. Authors are strongly encouraged to adhere to PLOS ONE writing style format (details on how to prepare the manuscript are available with the PLOS ONE Submission guidelines)

#2. The statistical reporting as well as table format should correspond to the requirements of PLOS ONE (see the Submission guidelines)

#3. The authors must adhere to the *Data Availability* policy of PLOS ONE.

#4. The data supporting the reported results should be available to any interested part without restrictions. The authors made available their data on a data access portal but with restrictions (e.g. any interested part will receive the dataset files within 48 hours). On top of that, the data files should allow the replication of *all* the results reported in the manuscript.

#5. While preparing their data files, authors should make sure that information on human subjects is properly anonymized. Details on preparing the dataset files are available on PLOS ONE webpage (see: *Data Availability* section).

#6. While assessing the final form of their manuscript, the authors are encouraged to follow PLOS ONE Submission Guidelines.

Reviewer #2: I understand the limitations associated with ethical approval and I am happy in this case with authors including data availability statement.

7. PLOS authors have the option to publish the peer review history of their article (what does this mean?). If published, this will include your full peer review and any attached files.

Reviewer #1: No

Reviewer #2: **Yes: **Karolina Urbanska

---

## [Author Response · Author response to Decision Letter 1]

6 Aug 2020

Our Cover letter details our response to reviewer and editor comments. In summary: 

We have revisited PLOS One’s submission guidelines and made the following improvements to the manuscript:

1. Minor improvements to readability

2. A leading zero has been inserted for all decimal numbers less than one. 

3. Use of square vs round brackets for citations in the text has been corrected

4. Minor formatting errors in the reference list have been corrected.

5. Vertical gridlines and single spacing has been applied to the tables.

6. The “Analyses” subsection of the Method is now labelled “Statistical Analysis” and contains additional information regarding the statistical procedures that were employed and our data data screening process.

---

## [Editor Report · Decision Letter 2]

10 Aug 2020

Research data sharing in the Australian national science agency: Understanding the relative importance of organisational, disciplinary and domain-specific influences

PONE-D-20-07169R2

Dear Dr. Mason,

We’re pleased to inform you that your manuscript has been judged scientifically suitable for publication and will be formally accepted for publication once it meets all outstanding technical requirements.

Kind regards,

Laurentiu Rozylowicz, Ph.D.

Academic Editor

PLOS ONE
---

## [Editor Report · Acceptance letter]

17 Aug 2020

PONE-D-20-07169R2 

Research data sharing in the Australian national science agency: Understanding the relative importance of organisational, disciplinary and domain-specific influences 

Dear Dr. Mason:

I'm pleased to inform you that your manuscript has been deemed suitable for publication in PLOS ONE. Congratulations! Your manuscript is now with our production department. 

Kind regards, 

on behalf of

Dr. Laurentiu Rozylowicz 

Academic Editor

PLOS ONE